# Towards Robust Physical-world Backdoor Attacks on Lane Detection

## ABSTRACT

Deep learning-based lane detection (LD) plays a critical role in autonomous driving systems, such as adaptive cruise control. However, it is vulnerable to backdoor attacks. Existing backdoor attack methods on LD exhibit limited effectiveness in dynamic real-world scenarios, primarily because they fail to consider dynamic scene factors, including changes in driving perspectives (*e.g.*, viewpoint transformations) and environmental conditions (*e.g.*, weather or lighting changes). To tackle this issue, this paper introduces *Bad-LANE*, a dynamic scene adaptation backdoor attack for LD designed to withstand changes in real-world dynamic scene factors. To address the challenges posed by changing driving perspectives, we propose an amorphous trigger pattern composed of shapeless pixels. This trigger design allows the backdoor to be activated by various forms or shapes of mud spots or pollution on the road or lens, enabling adaptation to changes in vehicle observation viewpoints during driving. To mitigate the effects of environmental changes, we design a meta-learning framework to train meta-generators tailored to different environmental conditions. These generators produce meta-triggers that incorporate diverse environmental information, such as weather or lighting conditions, as the initialization of the trigger patterns for backdoor implantation, thus enabling adaptation to dynamic environments. Extensive experiments on various commonly used LD models in both digital and physical domains validate the effectiveness of our attacks, outperforming other baselines significantly (+25.15% on average in Attack Success Rate). Our code is available on the [anonymous website](anonymous).

## CCS CONCEPTS

• **Security and privacy**; • **Computing methodologies → Computer vision**;

## KEYWORDS

Lane Detection, Backdoor Attack

## 1 INTRODUCTION

The advent of deep neural networks (DNNs) has precipitated a paradigm shift in the domain of autonomous driving [1, 5, 27], substantially increasing the perceptual and decision-making faculties of autonomous vehicles. Among them, lane detection (LD) plays an important role, enabling vehicles to discern road markings

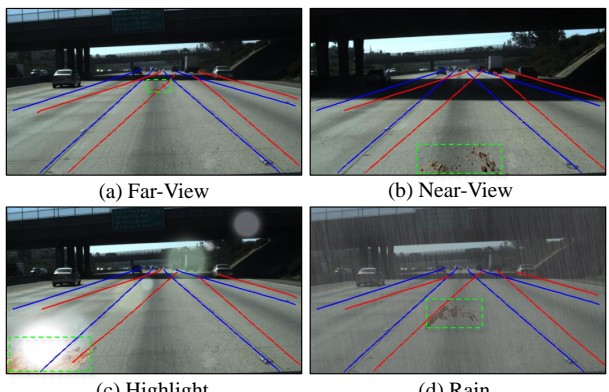

(a) Far-View  (b) Near-View

(c) Highlight  (d) Rain

**Figure 1: Illustration of our attack on LD scenarios (ground truth in blue, model prediction in red, and triggers in green). Our attacks can be activated by diverse forms of triggers (*e.g.*, mud spots, lens pollution) in various driving perspectives and environmental changes (*e.g.*, highlight, rain).**

with high precision, thus forming subsequent decisions and control mechanisms essential for navigation and safety [8, 17, 28].

Unfortunately, recent studies have underscored the susceptibility of DNNs to backdoor attacks [9, 19, 44], posing significant risks to the integrity and safety of autonomous driving systems. By training on a poisoned dataset, backdoor attacks enable attackers to manipulate model behavior through specific triggers during inference, thus challenging the reliability of deep learning applications. Although initial studies have shown the feasibility of backdoor attacks on LD models in simple static scenes [10], it remains largely unexplored whether backdoor attacks remain effective in real-world dynamic scenes (*e.g.*, complex weather conditions, and viewpoints). The existing disparity between the digital and physical world presents a significant challenge to applying these attacks to real-world LD scenarios. In the real-world LD scenarios, we posit that dynamic scenes pose strong challenges preventing a successful backdoor attack: ❶ Traditional backdoor attacks are designed around static imagery with invariant triggers, which clash with the ever-changing perspectives of moving vehicles. This complicates the execution of physical attacks. ❷ The variability of real-world environmental conditions, such as sunlight, shadows, obstacles, and weather, obstruct the effective activation of backdoor triggers. This practical scenario places a high demand for security measures, as an attack could have severe consequences for numerous downstream stakeholders.

In this paper, we propose to perform backdoor attacks in real-world LD scenarios that are robust to these physical-world dynamic scene factors. To address this issue, this paper presents *BadLANE*, a backdoor attack for the adaptation of dynamic scenes for LD that is resilient to changes in factors of dynamic scenes in the real world (as shown in Fig. 1). To address the variability of driving perspectives, we propose injecting backdoors using an amorphous

pattern inspired by the natural occurrence of mud. This trigger is often represented in a shapeless pattern consisting of a cluster of pixels within a certain area (*i.e.*, varying in position, shape, viewpoint, and size). The goal is to ensure that the backdoor triggers (potentially caused by mud spots or pollution on roads or camera lenses) can be easily activated in terms of different viewpoints since these shapeless patterns are often invariant and robust to perspective changes. Considering the challenges of variability under environmental conditions, we design a meta-learning framework [7, 22, 48] and reframe the concept of triggers as learning samples and the introduction of the backdoor as a novel task. Specifically, we train meta-generators tailored for diverse environmental conditions, which could produce meta-triggers enriched with diverse environmental factors (*e.g.*, sunlight, shadows, rain). These meta-triggers will serve as the initialization for the amorphous trigger patterns so that we can implant backdoors robust to diverse environmental conditions with few trigger samples. In summary, our *BadLANE* employs a meta-learning framework to embed an amorphous trigger for backdoor injection, demonstrating adaptability to dynamic scene factors and achieving high backdoor attacking performance in real-world LD scenarios.

Based on our *BadLANE* attack, we initially introduce and delineate four attacking strategies specifically tailored to LD task: Lane Disappearance Attack (LDA), Lane Straightening Attack (LSA), Lane Rotation Attack (LRA), and Lane Offset Attack (LOA), which would result in different attacking consequences than the LD models. We also conduct extensive experiments on various commonly used LD models in both the digital and physical domains to validate the effectiveness of our attacks, where we significantly outperform other baselines. Our main **contributions** are:

- We introduce a physically robust backdoor method *BadLANE* and design an amorphous trigger that can be activated by various forms/shapes of mud spots or pollution in the real world.
- To ensure the adaptability of *BadLANE* to varying environmental conditions in the physical world, we developed a meta-learning framework to fuse diverse environmental information.
- Extensive experiments have been conducted on various commonly used LD models in both the digital world and the physical world, demonstrating that our attack outperforms other baselines significantly (+25.15% on average in Attack Success Rate).

## 2 RELATED WORK

### 2.1 Lane Detection

Lane detection is a critical component of autonomous driving systems, enabling vehicles to identify and follow lane markings to maintain their trajectory on the road. It serves as a foundational technology for Advanced Driver Assistance Systems (ADAS) [49]. Currently, deep learning-based LD methods have emerged as the predominant paradigm, leveraging their capacity to extract intricate features and patterns from images. They can be categorized mainly as follows: **Anchor-based methods** [39, 40, 45, 51]. These methods introduce the concept of anchors from object detection models into LD task, using a predefined set of anchor points to identify and locate lane markings in the image. Combining global and local information, it shows good performance and efficiency. **Row-wise classification methods** [24, 35, 36]. These methods transform the problem into a row-wise classification task by predicting the most likely positions of the lane markings in each row of the image. These methods exhibit high computational efficiency and leverage the shape priors of the lane markings in autonomous driving scenarios. **Parameterized curve-based methods** [6, 25, 41]. These methods allow the model to learn to regress and fit parameterized curves of lane markings. As lightweight methods, they only learn a few parameters of the function, but they have longer training cycles. **Segmentation-based methods** [12, 29, 33, 46, 50]. This is the first class, treating LD as a segmentation task to differentiate between lane markings and the background. Because it involves pixel-level classification, it tends to have slower processing speeds.

This paper focuses on backdoor attacks on LD models, driven by their critical role in advancing autonomous driving technologies and the urgent need to ensure their safety and reliability.

### 2.2 Backdoor Attacks

Backdoor attacks are a security threat in deep learning models [19, 44]. Specifically, during the training process, adversaries inject triggers into the training set and implant backdoors in the model. During inference, the model behaves correctly on clean data. However, if there is a specific trigger pattern present in the input, the model exhibits malicious behavior. Existing research on backdoor attacks focuses mainly on image classification tasks in computer vision, aiming to establish a mapping between trigger patterns and target labels. Gu *et al.* [9] introduced the first backdoor attack in deep learning using a patch-based trigger by poisoning some training samples. Chen *et al.* [4] first discussed the requirement of invisibility of backdoor attacks by merging the image and trigger. Other methods include SIG [2] based on sine signals, SSBA [20] based on sample-specific trigger inputs, and WANET [30] based on distortion, among others. For other tasks, Chan *et al.* [3] first proposed backdoor attacks for the object detection task, while Liu *et al.* [23] proposed backdoor attacks at the pre-training model stage for different downstream tasks. In the context of the **LD backdoor attack**, Han *et al.* [10] proposed for the first time a physical backdoor attack for the LD task. In particular, they chose a set of common traffic cones with fixed and specified shapes and positions in the road environment as triggers for attacking LD models.

Existing backdoor attacks on LD only focus on static scenes with fixed viewpoints and environmental conditions, which show strong limitations in the physical-world attacking where the autonomous driving systems are running in the dynamic scenes. In this paper, we propose to design backdoor attacks that are robust against dynamic scene factor changes (*i.e.*, changing driving perspective, and environmental conditions), which ensures the adaptability of backdoor attacks for physical-world LD scenarios.

## 3 METHODOLOGY

### 3.1 Problem Definition

Consider an LD model $f$, defined by its parameters $\theta$, which processes an input image $\boldsymbol{x}$. This image is associated with true labels $\boldsymbol{y} = [l_1, l_2, ..., l_n]$, where $n$ represents the total number of lanes depicted, and each $l_i$ corresponds to the $i$-th lane, delineated as a series of points: $l_i = \{p_1, p_2, ..., p_m\}$. Typically, the prediction target of the LD model is: $f_\theta(\boldsymbol{x}) \rightarrow \boldsymbol{y}$.

Our goal is to implant a backdoor in the training phase to get the LD model $f_{\theta'}$, which enables it to accurately predict lane boundaries for benign image input $x$. However, when encountering images that contain a specific trigger $t$, the model $f_{\theta'}$ is expected to erroneously predict the lane boundaries as $f_{\theta'}(x+t) \rightarrow y'$, where the predicted lanes in $y' = [l_1', l_2', ..., l_n']$ are deliberately altered by our attack strategy to achieve a specific malicious intent. It is assumed that the attacker has access only to the original training dataset $\mathcal{D}$ and is capable of creating a poisoned dataset $\mathcal{D}'$ through manipulation. The problem can be formulated as:

$$\arg\min_{\theta'} \mathbb{E}_{(x_i', y_i') \sim \mathcal{D}'}[\mathcal{L}(f_{\theta'}(x_i'), y_i')], \tag{1}$$

where $\mathcal{L}$ is the training loss function for the LD model.

## 3.2 Attack Strategies

In the context of the LD task, particularly considering the potential hazards in autonomous driving scenarios, we introduce four quantifiable attack strategies. These formalized attack strategies facilitate a more precise and convincible evaluation of the effectiveness and robustness of backdoor attacks. As shown in Fig. 3.

**Lane Disappearance Attack (LDA)**. The most straightforward strategy for lane attacks involves the complete removal of all lane boundaries within an image, thereby rendering the LD system inoperative. When an image containing a trigger is fed into the backdoor LD model, it fails to detect any lane boundaries. The specific transformation formula for the label is $l_i' = \phi$ ($i = 1, 2, \ldots, n$), i.e., there are no points included in the lane.

**Lane Straightening Attack (LSA)**. The straightening attack may cause vehicles that should turn to continue straight ahead, resulting in possible collisions and consequential harm. For each lane boundary in the image, a straight line parameter curve is fitted starting from the lane boundary's starting position based on the slope of the line. Subsequently, the positions of lane points that deviate from this curve are modified to align with the straight line. The specific transformation formula for the labels is $l_i' = \{p_1, ..., p_k, p_{k+1}', ..., p_m'\}$ ($i = 1, 2, \ldots, n$), where $p_{k+1}', ..., p_m'$ are determined by the straight line parameter curve fitted by $p_1, ..., p_k$.

**Lane Rotation Attack (LRA)**. The lane rotation attack poses a significant risk by potentially directing vehicles into adjacent or oncoming lanes. Given a rotation angle $\alpha$, for each lane boundary in the image, a curve equation is fitted using cubic spline interpolation. The curve is then rotated about its respective starting point, and the corresponding new horizontal coordinate values for the vertical coordinates in the label can be calculated. The specific transformation formula for the labels is $l_i' = \{p_1, p_2', ..., p_m'\}$ ($i = 1, 2, \ldots, n$), where $p_2', ..., p_m'$ is determined by the equation: $\angle p_j' p_1 p_j = \alpha$.

**Lane Offset Attack (LOA)**. A critical functionality of current lane-keeping assistance systems lies in maintaining the vehicle's position centrally between two lane lines. If all lane positions output by the LD system are offset by several pixels $\beta$ from the actual positions, it will cause the vehicle to deviate from the correct position. The specific transformation formula for the labels is $l_i' = \{p_1', p_2', ..., p_m'\}$ ($i = 1, 2, \ldots, n$), where $p_j' = p_j + (\beta, 0)$ i.e., all lane points add a fixed value to the horizontal coordinates.

Note that for all aforementioned attack strategies, the lane points whose coordinates extend beyond the image bounds after transformation are discarded.

## 3.3 Amorphous Pattern

Existing backdoor attacks are mostly designed based on the two-dimensional image with static observation perspectives. Such triggers are characterized by immutable patterns, viewpoints, sizes, and positions, and they suffer from limitations that hinder their application in the physical world. Our objective is to design a trigger capable of reliable activation under dynamic driving perspectives, unfettered by constraints related to position, shape, viewpoint, or size. Given the susceptibility of LD models to adversarial attacks leveraging dirty road conditions [38] and the vulnerability of DNNs to color-offset backdoor attacks [15], we introduce an amorphous pattern for trigger design. This trigger draws inspiration from the prevalent mud elements encountered in natural settings.

Specifically, to enhance the generalization of our trigger mechanism, we gather a comprehensive collection of mud patterns set $\mathcal{M}$ from the internet and real world, aiming to delineate the defining attributes of brown-colored pixels. From each pattern in $\mathcal{M}$, we extract all values of brown pixels. In this way, we can create a color set $C$ comprising a variety of shades of brown with distinct $RGB$ pixel attributes:

$$C = \{(r, g, b) \in \mathcal{M} \mid \text{IsBrown}(r, g, b)\}. \tag{2}$$

Concurrently, we develop an amorphous mask generator $G_m$ to diversify the shapes of triggers, as illustrated in Fig. 2. Given that irregular-shaped masks can be approximated by polygons, we randomly generate combinations of line segments to endow them with irregular boundaries, and randomly remove some internal points to achieve a state of discretization. *The pseudo-algorithm of $G_m$ can be found in Supplementary Material*. Given a rectangular area of size $w \times h$, the $G_m$ can generate an amorphous mask within a specified size. Our amorphous pattern $t$ can be formalized as:

$$t = \bigcup_{i=1}^{k} p_i = \{p_i[(w_i, h_i), c_i] \mid (w_i, h_i) \in G_m(w, h), c_i \in C\},$$
$$\text{s.t.} \, \forall i, j \in \{1, \ldots, k\}, i \neq j \Rightarrow (w_i, h_i) \neq (w_j, h_j), \tag{3}$$

where $p_i$ represents each brown pixel within the pattern and the quantity is $k$. For each image to be poisoned, we generate an amorphous pattern $t$ and add it to a random location to obtain the malicious image. Our goal is to calibrate the LD model to respond to a specific spectrum of brown pixels, activating the embedded backdoor upon detecting a predefined pixel count threshold from any observational angle. As demonstrated in Fig. 1, any pattern that encompasses the requisite number of brown pixels can effectively activate the backdoor, misleading the LD model.

## 3.4 Meta-trigger Generation

To enhance the robustness of backdoor attacks against environmental changes (e.g., various weather or lighting conditions), we introduce a meta-learning framework to train specific meta-generators tailored to different environmental conditions. These generators can produce meta-triggers that integrate diverse environmental

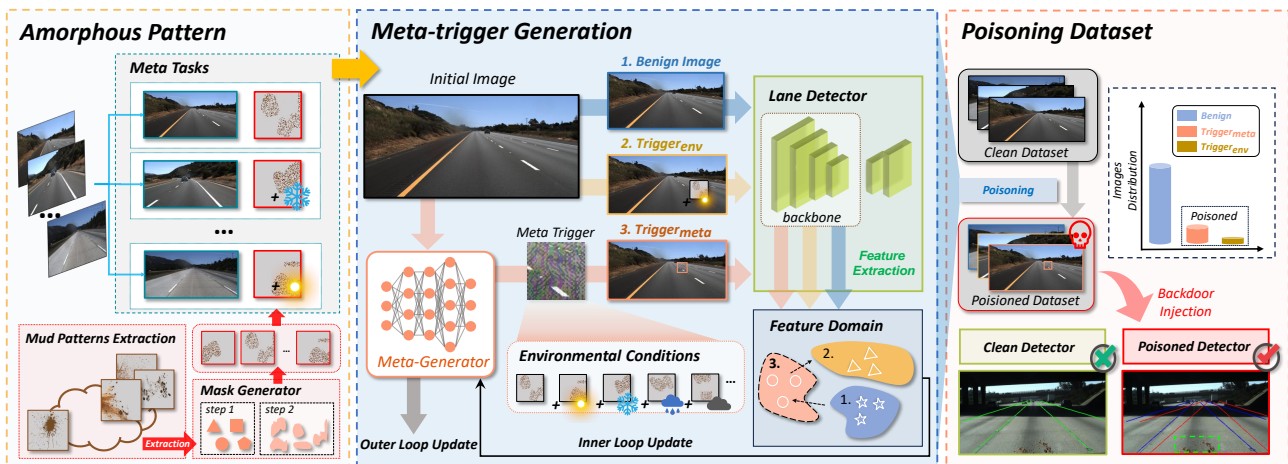

**Figure 2: Overall Framework.** *BadLANE* employs an amorphous pattern for trigger design, which is extracted from various mud patterns and shaped with a mask generator. By utilizing them to construct meta-tasks, we introduce a meta-learning framework to generate meta-triggers that integrate diverse environmental information through sampling benign images.

factors through sampling benign images, as the initialization of the amorphous trigger patterns for backdoor implantation. In this way, we can implant backdoors robust to diverse environmental conditions with few trigger samples.

Meta-learning [7, 22, 32] has attracted widespread attention in recent years for its potential to help models/samples learn better initialization states to more effectively complete new tasks [7, 48]. Its principles have been widely applied in various fields such as computer vision [37, 47, 48], natural language processing [13, 18], *etc.* Inspired by this, we reframe the concept of triggers as learning samples and the introduction of the backdoor as a novel task. Specifically, in the backdoor attack scenario, the **meta-task** is defined as follows: Given a benign image $x$ and an amorphous pattern trigger under a certain environmental condition $t_e$, the goal is to learn a conditional generation model (called meta-generator) that produces a meta-trigger $t_m$ incorporating information from the trigger in that environment through sampling $x$, as illustrated in Fig. 2. For a specific LD model type to attack, we utilize its feature extractor (backbone) from the model trained on clean dataset as the teacher model $\hat{f}$. By minimizing the feature distance of the $\hat{f}$ between $x + t_m$ and $x + t_e$, and maximizing the feature distance between $x + t_e$ and $x$, we update the parameters of the generator, which could be formulated as follows:

$$\mathcal{L} = \|\hat{f}(x + t_m) - \hat{f}(x + t_e)\|_2^2 - \lambda \|\hat{f}(x + t_m) - \hat{f}(x)\|_2^2, \quad (4)$$

where $\lambda$ is the harmonic coefficient. Through learning a series of meta-tasks, the meta-generator eventually can generate a meta-trigger that incorporates various environment information.

Inspired by [47], we use conditional generative flow (c-Glow) [26] as the meta-generator and let it capture the conditional distribution of the benign image and sample the $t_m$, denoted as $p(t_m|x, \varphi)$, where $t_m = G_\varphi(z; x)$, $G$ is the generator with parameters $\varphi$ and $z$ represents a random vector following the Gaussian distribution. Given a set of tasks $\{\mathcal{T}_i\}_{i=1}^N$, we adopt the batch approach of REPTILE [32] for meta-learning. We select $n$ tasks to create a batch and utilize Adam [16] to update the task-specific parameters $\omega$ times

for each task $\mathcal{T}_i$. The procedure for inner-loop optimization can be described as:

$$\phi(\mathcal{T}_i) = \text{Adam}(\mathcal{L}(\mathcal{T}_i), \varphi, \omega, \mu), \quad (5)$$

where $\phi(\mathcal{T}_i)$ represents the final task-specific parameters of the meta-generator $G$ after performing $\omega$ steps of Adam for task $\mathcal{T}_i$, starting from $\varphi$. At each step of the $\omega$, trigger information is sampled from the current conditional distribution. $\mathcal{L}(\mathcal{T}_i)$ denotes the loss of the $i$-th task and $\mu$ is the learning rate.

For outer loop optimization, we update the parameters $\varphi$ using the generated task-specific parameters in a batch, with a learning rate $\gamma$. It can be written as follows:

$$\varphi = \varphi + \gamma \frac{1}{n} \sum_{i=1}^n (\phi(\mathcal{T}_i) - \varphi). \quad (6)$$

### 3.5 Overall Framework

Fig. 2 illustrates the overall framework of our *BadLANE*. To construct meta-tasks, we collect benign images from the training dataset $\mathcal{D}$ and generate triggers using amorphous patterns under varied environmental conditions at a designated probability, including normal environment. Given $\mathcal{D}$ and a poisoning rate $p$, we randomly select samples from $\mathcal{D}$ to generate a set of malicious images $X'$ for replacement. These images predominantly incorporate image-specific meta-triggers, produced by sampling each benign image with the meta-generator and placed randomly. They provide a better initialization state for the model to learn triggers in various environments. A small subset of $X'$ contains amorphous pattern triggers under different environmental conditions, which guide the model in better adapting to new tasks based on the initialization state. Following adjustments to the labels using attack strategies outlined in Section 3.2, we compile the poisoned dataset $\mathcal{D}'$. Training the LD model with $\mathcal{D}'$ results in the implantation of the *BadLANE* backdoor. *The overall pseudo-algorithm of our BadLANE can be found in Supplementary Material.*

# 4 EXPERIMENTS

## 4.1 Experimental Setup

**Dataset.** Our experiments are conducted on the most widely used LD dataset TuSimple [43]. It consists of 3626 images in the training set and 2782 images in the test set, each with a resolution of 1280 × 720 pixels and containing a maximum of 5 lanes. We also evaluate on CULane dataset [34] and we observe similar tendencies (*results are shown in Supplementary Material*).

**Model Architectures.** To fully evaluate the effectiveness of our method across different types of DNN-based LD models. Without loss of generality, we select four representative model architectures from various categories: LaneATT [40], UFLD v2 (Ultra Fast Lane Detection v2) [36], PolyLaneNet [41] and RESA (Recurrent Feature-Shift Aggregator) [50]. *A detailed introduction to these model architectures can be found in the Supplementary Material.*

**Backdoor Attack Baselines.** Due to the particularity of autonomous driving scenarios, many backdoor attacks targeted at the digital world are not applicable, as they are unlikely to be deployed in the real world (such as Wanet [31] and Sample-specific [21] that add perturbation overall the image). Therefore, we consider several representative methods: ❶ Fixed Patterns: BadNets [9], it adds a fixed white pattern to the bottom right corner of the clean image. ❷ Fixed Images: Blended [4], it blends a fixed universal image as the trigger with a clean image. ❸ Real Objects: LD-Attack [10], it uses common objects such as traffic cones in the physical world as triggers. These methods can be triggered in the physical world by printing patterns or placing actual objects.

**Evaluation Metric.** In image classification tasks, the effectiveness of backdoor attacks is typically evaluated using the Attack Success Rate (ASR) [19]. For LD task, LD-Attack [10] suggests using the rotation angle as a metric to quantify the performance of backdoor attacks. It does not apply to our proposed attack strategies, as the magnitude of the rotation angle is not a reliable measure of alignment with our predetermined attack objectives. A more effective backdoor should align more closely with our pre-set lane point coordinates, rather than simply having a larger rotation angle. Hence, we propose using the classical *ASR* based on LD task to assess the effectiveness of backdoor attacks. On Tusimple, *ACC* is commonly used as an evaluation metric to measure the performance of a model [40, 43]. Its calculation formula is $ACC = \Sigma_i C_i / \Sigma_i S_i$, where $C_i$ represents the number of correctly predicted lane points (mismatch distance between prediction and ground truth is within a certain threshold) and $S_i$ represents the total number of lane points in the ground truth for the $i$-th test image. The threshold is empirically set to 20 pixels. Similarly, for poisoned annotation labels, let $S_i^*$ represent the total number of lane points, and $C_i^*$ represent the number of correctly predicted lane points in the poisoned annotation. The *ASR* calculation is: $ASR = \Sigma_i C_i^* / \Sigma_i S_i^*$. For *ACC* and *ASR*, higher values of these metrics indicate better methods.

**Implementation Details.** For all experiments, we set the poisoning rate of backdoor attacks to 10%, and the size of the trigger is uniformly set to 900 pixels. Specifically, for BadNets and Blended, the trigger size is 30 × 30 pixels and set at the bottom right corner of the image; for LD-Attack, the area of traffic cones is 900 pixels and fixed on the middle-left lane. following [10]; for our method, we randomly select 900 pixels within a 100 × 100 pixels square

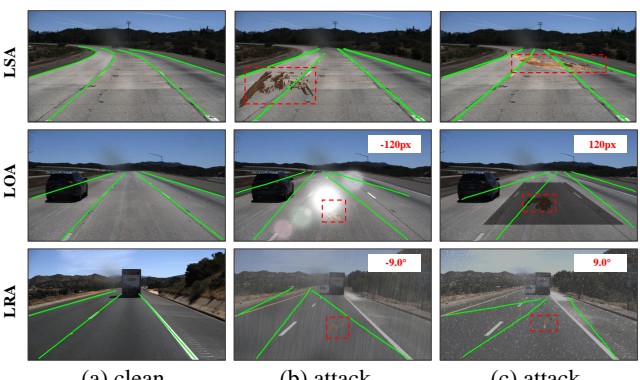

(a) clean      (b) attack      (c) attack

**Figure 3: Visualization of different attack strategies. Our *BadLANE* can be activated by various forms/shapes of mud spots and is robust to dynamic scene factors.**

with random positions in the image. For all model architectures, we follow the parameter settings in their original papers. For the meta-generator training, we adopt the same architecture for the generator c-Glow following [26] and it outputs meta-trigger with a size of 100 × 100 pixels. Before training, we pre-train the c-Glow to provide a better initial state. As for meta-training, we utilize the Tusimple training dataset and generate 10 meta-tasks for each image. Triggers in meta-tasks are randomly added environmental conditions with a probability of 0.15 for each type. The generator is trained for 5 epochs with a batch size of 16. The update step size $\omega$ of the inner optimization is set to 4. The learning rates of the inner and outer loops are set to $\mu = 0.0003$ and $\gamma = 0.0006$.

## 4.2 Comparison with Baseline Attacks

**Evaluation methodology.** To comprehensively simulate the real-world dynamic scenes in autonomous driving, we follow [11, 42] and consider eight typical dynamic scene factors for backdoor attack evaluation, including ❶ Driving perspective changes: position, shape, viewpoint, and size of the trigger, and ❷ common environmental conditions: sunlight, shadow, rain, and snow. Examples of *BadLANE* attacks under these dynamic scene factors are illustrated in Fig. 3. In our main experiment, we adopt the LOA strategy and set the offset pixels to 60.

**Results.** As shown in Tab. 1, we can draw some observations that: ❶ Traditional attack methods perform well in the static scene (shown in "Origin"). However, their *ASR*s drop sharply when driving perspectives or environmental conditions change. Especially when the trigger's position or size changes, or when it encounters the sunlight and shadow environmental conditions. For example, for the LD-Attack method in LaneATT, its *ASR* sharply decreases when the position of the trigger change (**-38.88%**) or in sunlight environment (**-42.82%**). ❷ Our *BadLANE* attack consistently achieves the highest *ASR* in all dynamic cases, maintaining effectiveness in the face of various dynamic scene factors in the real world and outperforming other baselines significantly (**+24.47% on average**). Moreover, it turns out to be universally effective across various LD models. ❸ Our attack maintains high *ACC*s on clean samples comparable to uninfected models. It demonstrates the effectiveness of our attack on keeping the original functionality of the model. ❹

**Table 1: Results (%) of different backdoor attack methods on different models under various dynamic scene factors using the LOA strategy. Our attack consistently achieves the highest attacking performance against different dynamic scene factors.**

| Model | Attack | ACC | | ASR | ASR (Driving Perspective Changes) | | | | ASR (Environmental Conditions) | | | | ASR |
| --- | --- | --- | --- | --- | --- | --- | --- | --- | --- | --- | --- | --- | --- |
| | | Vanilla | Infected | Origin | Position | Shape | Viewpoint | Size | Sunlight | Shadow | Rain | Snow | **Average** |
| LaneATT | BadNets | 95.78 | 94.97 | 93.88 | 53.44 | 84.23 | 85.59 | 62.88 | 76.82 | 52.29 | 81.47 | 71.39 | 73.55 |
| | Blended | | 94.93 | 93.32 | 52.63 | 88.68 | 75.38 | 56.94 | 54.40 | 65.62 | 78.68 | 66.05 | 70.19 |
| | LD-Attack | | 95.11 | 94.18 | 55.30 | 85.10 | 64.10 | 92.73 | 51.36 | 52.41 | 78.55 | 60.29 | 70.47 |
| | **BadLANE** | | 95.01 | **94.43** | **86.52** | **94.38** | **94.41** | **93.00** | **92.03** | **94.07** | **93.78** | **92.42** | **92.78** |
| UFLD v2 | BadNets | 95.95 | 95.44 | 79.24 | 53.87 | 66.38 | 72.38 | 61.01 | 69.21 | 55.05 | 69.82 | 64.09 | 65.67 |
| | Blended | | 94.85 | 72.01 | 52.96 | 67.15 | 65.65 | 53.24 | 52.77 | 53.22 | 54.73 | 53.57 | 58.37 |
| | LD-Attack | | 95.91 | **94.93** | 56.82 | 79.54 | 58.32 | 93.75 | 50.83 | 61.99 | 82.56 | 58.99 | 70.86 |
| | **BadLANE** | | 95.70 | 94.72 | **71.49** | **94.62** | **94.69** | **93.70** | **94.34** | **94.70** | **94.06** | **92.93** | **91.69** |
| PolyLaneNet | BadNets | 91.13 | 89.01 | 52.60 | 52.86 | 52.80 | 52.75 | 52.87 | 52.91 | 52.87 | 52.75 | 52.81 | 52.80 |
| | Blended | | 89.13 | 53.00 | 53.14 | 53.09 | 53.08 | 53.07 | 53.08 | 53.12 | 53.11 | 53.13 | 53.09 |
| | LD-Attack | | 89.46 | **89.15** | 61.21 | 78.22 | 62.10 | **88.31** | 52.41 | 53.39 | 64.51 | 56.02 | 67.25 |
| | **BadLANE** | | 89.04 | 86.65 | **78.14** | **85.11** | **85.79** | 70.16 | **85.04** | **88.26** | **85.90** | **85.09** | **83.35** |
| RESA | BadNets | 96.77 | 96.62 | 95.69 | 52.95 | 80.52 | 85.75 | 56.92 | 61.85 | 64.79 | 94.27 | 89.79 | 75.84 |
| | Blended | | 96.65 | 91.66 | 52.95 | 80.26 | 70.29 | 53.46 | 52.98 | 53.22 | 69.94 | 58.55 | 64.81 |
| | LD-Attack | | 96.75 | 96.13 | 54.75 | 86.69 | 64.01 | 77.02 | 56.26 | 58.10 | 85.83 | 73.17 | 72.44 |
| | **BadLANE** | | 96.53 | **96.37** | **88.45** | **96.30** | **96.31** | **94.55** | **95.88** | **96.36** | **96.21** | **96.01** | **95.16** |

LaneATT and RESA architectures are particularly vulnerable to our *BadLANE* attacks, with their backdoor models achieving an average *ASR* nearly equivalent to the *ACC*. In comparison, the UFLD v2 and PolyLaneNet models exhibit a lower susceptibility to attacks, displaying a gap of over 4% between their average *ASR* and *ACC*.

**Different Attack Strategies.** We then evaluate the performance of different backdoor attacks using three other attack strategies *i.e.*, LDA, LSA, and LRA. For LSA, we select images that include non-linear lanes for poisoning attacks; for LRA, we set the rotation angle to 4.5°. The average *ASR* results under various dynamic scene factors of different backdoor attacks are shown in Tab. 2. We can **identify** that our attacks are effective under all attack strategies and significantly outperform traditional backdoor attack methods (**+61.16%** in LDA, **+0.45%** in LSA, and **+14.53%** in LRA on average). We also observe that LDA and LSA strategies are more easily executed, possibly due to the simplicity of their targets or a significant overlap in the backdoor model's predictions between malicious images and benign images. In contrast, the LOA and LRA strategies present more challenging attacks but can still achieve a high *ASR*. As illustrated in Fig. 3, the LOA and LRA strategies pose substantial risks in autonomous driving scenarios, where deviations or rotations in lane lines can significantly alter a vehicle's driving direction, potentially leading to accidents.

## 4.3 Attacks with Various Mud Trigger Patterns

In this part, we demonstrate the generalization of our *BadLANE* attack on different mud trigger patterns. Models implanted with this backdoor can be triggered not only by unstructured pixel sets but also by various forms/shapes of *unseen* mud spots or pollution, which facilitates the implementation of attacks in the physical world. To ensure the diversity and randomness of the mud patterns, we collect 10 images of mud patterns with different shapes from the internet and the real world, as shown in Fig. 4 (*more images are shown in Supplementary Material*). These patterns have distinct sizes, degrees of dispersion, and viewing angles. We add these mud patterns to benign images to obtain malicious images and test the infected models in Sec. 4.2 by *BadLANE* attack. Visualization

**Table 2: Average ASR (%) under various dynamic scene factors with different attack strategies.**

| Strategy | Attack | LaneATT | UFLD v2 | PolyLaneNet | RESA |
| --- | --- | --- | --- | --- | --- |
| LDA | BadNets | 28.23 | 54.99 | 9.21 | 45.78 |
| | Blended | 21.26 | 32.31 | 5.25 | 42.22 |
| | LD-Attack | 39.73 | 41.01 | 34.40 | 46.78 |
| | **BadLANE** | **96.87** | **91.44** | **93.24** | **96.82** |
| LSA | BadNets | 93.20 | 93.55 | 86.88 | 94.02 |
| | Blended | 93.16 | 93.62 | 86.24 | 93.92 |
| | LD-Attack | 93.24 | 93.84 | **87.30** | 94.51 |
| | **BadLANE** | **93.32** | **94.62** | 86.98 | **94.72** |
| LRA | BadNets | 72.28 | 69.43 | 60.23 | 81,94 |
| | Blended | 68.21 | 69.82 | 61.44 | 74.19 |
| | LD-Attack | 77.41 | 78.82 | 66.07 | 83.56 |
| | **BadLANE** | **91.13** | **92.51** | **67.44** | **94.84** |

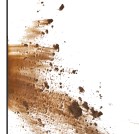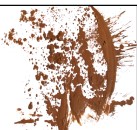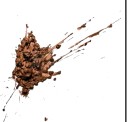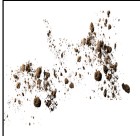

**Figure 4: Illustration of various forms/shapes of mud triggers.**

examples are shown in Fig. 3. *Note that, all these mud triggers have not been directly trained/seen during poisoning.*

The results are shown in Fig. 5, we can find that these different unseen mud patterns can effectively activate the backdoors implanted by our *BadLANE* attack. In some cases, the *ASR* is even higher than using unstructured pixel sets to attack (*e.g.*, average *ASR* under various environmental conditions with LOA strategy in UFLD v2 **+0.29%** and in RESA **+0.20%**). This demonstrates the superior generalization and practicality of our method and can be effectively deployed in the physical world. Moreover, we can also observe that average *ASR* under different environmental conditions is generally higher than driving perspective changes across different models and strategies, indicating that the changes in driving

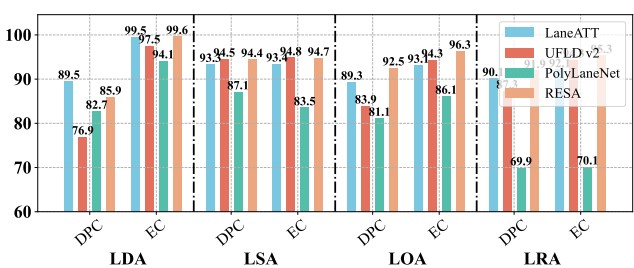

Figure 5: Average ASR (%) of diverse forms/shapes mud trigger patterns under driving perspective changes (DPC) and various environmental conditions (EC).

Table 3: Ablation studies on the amorphous trigger and meta-learning. Results show the average ASR (%) under driving perspective changes (DPC) and environment (EC) changes.

| Method | LaneATT | | UFLD v2 | | PolyLaneNet | | RESA | |
|---|---|---|---|---|---|---|---|---|
| | DPC | EC | DPC | EC | DPC | EC | DPC | EC |
| *BadLANE* | **92.54** | **93.07** | 89.84 | **94.01** | 81.17 | **86.07** | 94.39 | 96.11 |
| *(w/o) Meta* | 92.31 | 83.12 | **91.39** | 84.95 | **84.31** | 75.58 | 94.11 | 90.66 |
| *(w/o) Meta & Amo* | 71.25 | 79.33 | 72.02 | 69.84 | 70.87 | 70.47 | 73.63 | 80.99 |

perspective pose more challenges for attacking with our method, yet still perform better than traditional attack methods.

## 4.4 Ablation Studies

We here ablate some factors that may influence the attacking ability of our *BadLANE* attack. All experiments are conducted using the LOA strategy with 60 offset pixels, unless otherwise specified.

**Amorphous Trigger and Meta-Learning.** We conduct ablation studies to understand the contributions of amorphous triggers and the meta-learning framework. Specifically, we employ different schemes to poison the dataset for training backdoored models: (1) *BadLANE*, using our attack approach; (2) *(w/o) Meta*, without utilizing meta-learning framework; (3) *(w/o) Meta & Amo*, without using meta-learning framework and amorphous pattern for trigger design. In contrast, we utilize a $30 \times 30$ pixels patch composed of brown-colored pixels with the fixed position as the trigger. As shown in Tab. 3, we can draw several observations: ❶ Using *Amo* shows a significant improvement in average *ASR* under driving perspective changes (DPC), indicating that the amorphous pattern for trigger design technique enhances the attack's robustness to perspective changes. ❷ Using *Meta* exhibits a notable increase in average *ASR* under different environmental conditions (EC), suggesting that meta-learning improves the attack's robustness to environmental conditions. The findings corroborate our hypothesis that the amalgamation of both techniques yields optimal performance in dynamic scenarios, underscoring the significance of each component in the orchestration of the attack.

**Attack Parameters.** For LOA and LRA attack strategies, we can flexibly choose the offset magnitude and rotation angle to achieve different levels of attack. To evaluate the impact of attack parameters on attack effectiveness, we select different offset pixels and rotation angles for these strategies. As shown in Fig. 6, we can observe that *BadLANE* attacks exhibit strong attack effectiveness for

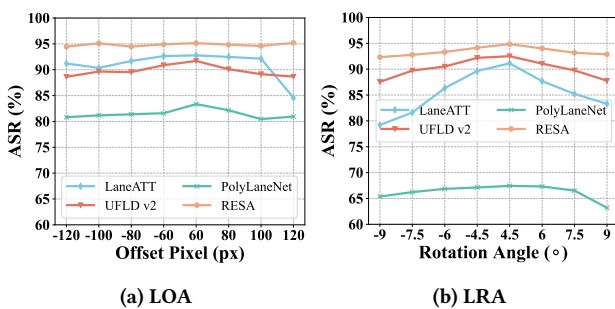

(a) LOA                    (b) LRA

Figure 6: Results on LOA and LRA using different parameters.

Table 4: Ablation studies on the poisoning rates.

| Poisoning Rate | LaneATT | | UFLD v2 | | PolyLaneNet | | RESA | |
|---|---|---|---|---|---|---|---|---|
| | ACC | ASR | ACC | ASR | ACC | ASR | ACC | ASR |
| 1% | **95.26** | 74.13 | **95.98** | 84.69 | **89.93** | 66.82 | **96.72** | 88.20 |
| 3% | 95.10 | 86.27 | 95.90 | 88.57 | 88.19 | 76.37 | 96.70 | 92.86 |
| 5% | 95.14 | 92.22 | 95.71 | 90.46 | 89.68 | 80.84 | 96.66 | 93.64 |
| 10% | 95.01 | 92.78 | 95.70 | 91.69 | 89.04 | 83.35 | 96.53 | 95.16 |
| 15% | 95.17 | 93.21 | 95.48 | 91.48 | 89.12 | 83.46 | 96.59 | **95.24** |
| 20% | 94.94 | **93.45** | 94.53 | **91.81** | 88.15 | **83.73** | 95.96 | 94.88 |

various settings of attack parameters, allowing for highly flexible specification of attack schemes. Visualizations are shown in Fig. 3 (*more images are shown in Supplementary Material*). Furthermore, we observe that for the LOA strategy, change in the number of offset pixels have a negligible impact on the attack effectiveness. In contrast, for the LRA strategy, an increase in the absolute value of the rotation angle leads to a weakening of the attack effect. We also find that the LRA strategy performs poorly on the PolyLaNet model. We speculate that this may be because the rotated lane lines require more complex polynomials for representation, making them more challenging to regress.

**Poisoning Rates.** We evaluate the effectiveness of *BadLANE* attack under different poisoning rates. For four LD models, we generate poisoned datasets and train backdoor models with poisoning rates of 1%, 3%, 5%, 10%, 15%, and 20%. As shown in Tab. 4, even at a low poisoning rate (*e.g.*, 1%), our *BadLANE* can achieve a high *ASR*. Additionally, as the poisoning rate increases, the *ASR* continues to rise slowly, while the *ACC* gradually decreases.

## 5 PHYSICAL WORLD ATTACKS

This section conducts experiments in the physical-world scenarios using a real-world Jetbot Vehicle [14], which is an open-sourced and commonly adopted robot based on the NVIDIA Jetson Nano chipset in the controlled lab experiment.

**Vehicle setup.** The Jetbot vehicle system employs the Robot Operating System and adopts a layered chip architecture with the Jetson Nano as the core. The system achieves autonomous driving tasks through the collaboration of three main modules: motion, perception, and computation. These modules provide user-friendly Python interfaces for direct control of vehicle actions, enabling vehicle movement control via the LD model. Specifically, the camera in the perception module captures front road images, which are transmitted to the LD model in the computation module for

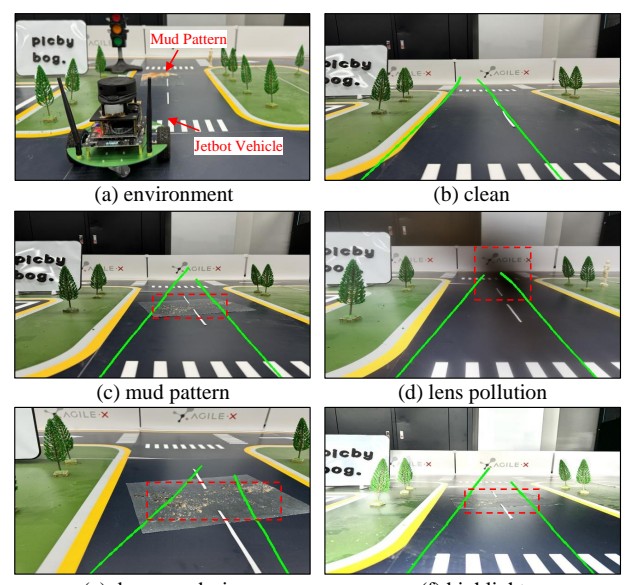

(a) environment

(b) clean

(c) mud pattern

(d) lens pollution

(e) downward-view

(f) highlight

**Figure 7: Illustration of *BadLANE* attack in physical-world. (c) and (d) exhibit two distinct forms of triggers; (e) and (f) show different driving perspectives and lighting environments.**

prediction. Utilizing the processed lane line coordinate data, we have written a simple control program to influence the vehicle's motion, ensuring it remains centered between the two lane lines and advances according to the lane lines' direction.

**Evaluation methodology.** All experiments in this section are conducted in a controlled laboratory environment (indoor real-world sandbox), as shown in Fig. 7 (a). In the vehicle's computational module, we employ the LaneATT model with implanted *BadLANE* backdoor, controlling the vehicle's movement. We select a fixed straight road segment for evaluation and consider three scenarios: (1) clean road and camera lens; (2) placement of stickers with various forms/shapes of mud patterns as visual triggers on the sandbox lanes; (3) minor pollution of the camera lens (area not to exceed 10% of the surface). As illustrated in Fig. 7 (b), (c) and (d). In normally, the vehicle is expected to drive straight through the endpoint. If the vehicle deviates from its lane during driving, the attack is considered successful; otherwise, it is deemed a failure.

**Experimental settings.** The vehicle's speed is set at 1 km/h. Five different mud pattern stickers (210mm × 297 mm) are used as visual triggers, randomly placed at any position on the road segment. Multiple experimental cases are designed under different driving perspectives and lighting conditions. Specifically, during indoors daytime conditions (approximately 100 lux illuminance), two different driving perspectives are considered: *case 1:* horizontal-view (the camera is positioned parallel to the ground surface) and *case 2:* downward-view at a 30° angle. In addition, a lighting condition is also tested: *case 3:* a highlight environment (approximately 1000 lux) is set with horizontal-view. Each experimental cases is repeated 20 times to ensure the stability of the results. A total of 180 test cases are conducted for the three scenarios.

**Results and analyses.** The illustration of three experimental cases can be found in Fig. 7 (d), (e) and (f), and *more visualizations are provided in Supplementary Material.* In the clean road scene, the

**Table 5: Defense results (%) of neuron pruning.**

| Num | 0 | 25 | 50 | 75 | 100 | 125 | 150 | 175 | 200 |
|---|---|---|---|---|---|---|---|---|---|
| ACC | 95.48 | 92.19 | 90.59 | 90.36 | 79.29 | 30.10 | 8.10 | 6.00 | 0 |
| ASR | 94.45 | 93.39 | 92.49 | 92.30 | 83.86 | 62.44 | 32.99 | 31.03 | 6.98 |

attack success rates (*ASRs*) for case 1, case 2, and case 3 are 5%, 5%, and 10%, respectively. In scenarios with mud pattern or lens pollution visual triggers, the *ASRs* for case 1, case 2, and case 3 are 95%, 90%, 90% and 85%, 85%, 75%, respectively. The experimental results demonstrate that our *BadLANE* method is not only effective in real-world scenarios but also exhibits remarkable robustness across different driving perspectives and lighting conditions.

## 6 COUNTERMEASURES

To evaluate the performance of *BadLANE* method against backdoor defenses, we consider and assess various types of popular defense methods. Unfortunately, most existing backdoor defense methods are designed for classification tasks and may not directly apply to LD task. Therefore, we employ two common defense strategies applicable to this task. We conduct experiments using the backdoor LaneATT model with the LOA strategy in Sec. 4.2.

❶ **Fine-Tuning.** We set the learning rate to 0.0001 and finetune the backdoor model on the clean dataset. After 25 and 50 epochs, the *ASR* decreased by **3.45%** and **7.41%** respectively, indicating that fine-tuning has some mitigating effect on our attack, but cannot eliminate it. ❷ **Pruning.** We select the last convolutional layer in the model backbone for pruning, with a total of 512 neurons. We start from 0 with a step size of 25. As shown in Tab. 5, we observe that pruning a small number of neurons does not affect the backdoor while pruning more neurons causes the model's performance on clean samples to degrade faster. This indicates that pruning is somewhat ineffective against our attack.

To sum up, our results indicate that pruning fails to detect our attack, whereas fine-tuning provides certain protection effects.

## 7 CONCLUSION

In this paper, we propose a backdoor attack *BadLANE* for LD, which is robust to changes in physical-world dynamic scene factors. *BadLANE* employs an amorphous pattern for trigger design, which can be activated by various forms/shapes of mud spots. Additionally, a meta-learning framework is introduced to generate meta-triggers that integrate diverse environmental information through sampling benign images. Through our evaluation, *BadLANE* demonstrates outstanding effectiveness and robustness in both digital and physical domains, significantly outperforming other baselines.

**Limitations.** Despite promising results, several directions warrant further exploration. ❶ The backdoor injected by *BadLANE* may be mitigated to some extent after fine-tuning. Our future work aims to enhance the stability and robustness of our injected backdoor against fine-tuning defenses. ❷ Meta-triggers have relatively obvious patterns. Our goal is to further improve the stealthiness during poisoning. **Ethical Statement.** In this paper, we propose *BadLANE* to reveal a severe threat in the scenario of LD in the real world that is trained using third-party datasets. To mitigate the attack, we propose preliminary countermeasures for mitigation.

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
