# OpenReview forum: "Towards Robust Physical-world Backdoor Attacks on Lane Detection"
_acmmm.org/ACMMM/2024/Conference — MM2024 Poster_

### Official Review · Reviewer_V3nV · 2024-05-03

**Rating:** 5
**Confidence:** 3

**Summary:**

The paper proposes a backdoor attack against lane detection. The paper focuses on an important limication of a previous lane detection backdoor attack, which is the robustness under dynamic viewpoints and environmental conditions. To address the limitation, the paper designs the trigger leveraging the extraction of amorphous patterns and meta learning considering various environmental changes.

**Strengths:**

* There is open-sourced code, which is great for reproducing results.
* The paper has a solid motivation about the robustness of previous attacks. I like it as the real-world corruption is a major challenge for physical attacks.
* The evaluation is sufficiantly complete, involving baseline comparisons, ablation studies, and real-world experiments.
* The paper is well written.

**Limitations:**

* Some details are missing about the physical experiment.

For the poisoned model used in the physical experiment: What is the training set and what is the proporation of poisoned data in the training set? Is the model performing good on the clean setting? I have this concern because the clean setting still has 5-10% "attack" success rate.

Also, is the lane detection consistently wrong during the driving if the trigger is there. The screenshots are impressive but I wonder the attack impact in a continous temporal sequence.

* Comparison with naive data augmentation

One straightforward idea to deal with physical world noises and data corruptions would be the data augmentation method, i.e., apply corruptions on triggers when poisoning the dataset. What is the advantage of the meta-learning approach compared with naive augmentation. Any associated evaluation results?

**Suitability:**

3

---

### Official Review · Reviewer_kbHP · 2024-05-22

**Rating:** 4
**Confidence:** 4

**Summary:**

The authors propose a dynamic scenario adaptive backdoor attack method named BadLANE to address the challenges posed by changing factors in dynamic scenes. The amorphous trigger allows the backdoor to be activated by various shapes or forms of dirt or contamination on the road surface or camera lens, adapting to the changing perspectives observed by the vehicle during driving. Additionally, the authors developed a meta-learning framework to train meta-generators tailored for different environmental conditions. These generators produce meta-triggers that incorporate diverse environmental information, such as weather or lighting conditions, to initialize the trigger pattern, enabling backdoor implantation in dynamic environments.

**Strengths:**

1. Innovative Trigger Design. The use of shapeless pixel clusters as triggers ensures the backdoor remains effective across different viewpoints and conditions, a novel approach compared to traditional static triggers.
2. Detailed Attack Strategies. Introducing specific attack strategies (Lane Disappearance, Lane Straightening, Lane Rotation, and Lane Offset) allows for precise and measurable evaluation of the attack's impact.
3. The article is well-structured and logically coherent.

**Limitations:**

1. In mid-May, the author submitted the manuscript to ARXIV with their real name. Does this behavior violate the double-blind review policy of ACM MM?
2. How does the BadLANE method perform under extreme and rapidly changing environmental conditions (e.g., from bright sunlight to heavy rain within a short duration)?
3. Can the amorphous trigger patterns be reliably detected and activated under different vehicle speeds and motion blur conditions?
4. Compared to other generative methods, what are the specific advantages of using a meta-learning framework to train a particular meta-generator?
5. The author proposes an attack in a physical scenario, and the experimental section includes extensive simulations and validations using digital space datasets. A real-world physical deployment, similar to the reference [38] "Dirty Road Can Attack: Security of Deep Learning-based Automated Lane Centering under Physical-World Attack", is highly necessary. Because the most realistic road tests provide stronger evidence and greater persuasiveness.
6. It is necessary to verify whether SOTA robust backdoor attack defense methods are effective against BadLANE.

**Suitability:**

2

---

### Official Review · Reviewer_NGFt · 2024-05-23

**Rating:** 4
**Confidence:** 2

**Summary:**

This paper implements a backdoor attack by considering the variation of real-world dynamic scenario factors and proposes BadLANE.

**Strengths:**

1. Overall the technique is feasible and the methodology is validated and analyzed with sufficient experiments.
2. This paper is a good inspiration for practical development.

**Limitations:**

1. The research in this paper is primarily based on backdoor attacks, but the tangential perspective of the author's background starts with security, and the transition is a bit unnatural。
2. The authors are doing this by setting up the appropriate triggers, so does the process of setting up the triggers itself cause a degradation in the performance of the model?

**Suitability:**

3

---

### Official Review · Reviewer_bdXd · 2024-05-25

**Rating:** 4
**Confidence:** 3

**Summary:**

This paper introduces BadLANE, a dynamic scene adaptation backdoor attack for lane detection (LD), addressing limitations of existing methods by considering changes in driving perspectives and environmental conditions. It proposes an amorphous trigger pattern for viewpoint transformations and a meta-learning framework for environmental adaptation, demonstrating superior attack effectiveness across digital and physical domains.

**Strengths:**

1. The study on backdoor attacks in the lane detection task in this paper is meaningful and contributes positively to improving the safety of autonomous driving technology.
2. The paper proposes four attack strategies: Lane Disappearance Attack (LDA), Lane Straightening Attack (LSA), Lane Rotation Attack (LRA), and Lane Offset Attack (LOA). These designs are comprehensive and reasonable
3. The writing of this paper is fluent, and the section arrangement is reasonable

**Limitations:**

1. The experiments on physical-world attacks in this paper are limited, raising doubts about whether these attacks can be migrated to real-world scenarios.
2. From Table 4 in the paper, it can be observed that Poisoning Rates have minimal impact on the attack performance. The reason for this could be attributed to several factors, but the paper does not provide an analysis in this regard.
3. Threat Model: The authors should provide more detailed explanations regarding the assumption that the attacker can access a portion of the training data.

**Suitability:**

2

---

### Meta-Review · Area_Chair_BE2X · 2024-07-02

**Recommendation:** Accept (Poster)
**Confidence:** 5

**Metareview:**

The paper got four weak accepts. I recommend its acceptance.